# Influence of Natural Deep Eutectic Solvent Compositions on the Polyphenol Profile of *Citrus aurantium* By-Products from Yucatán, México

**DOI:** 10.3390/molecules30234551

**Published:** 2025-11-26

**Authors:** Joaquín Fernández-Cabal, Kevin Alejandro Avilés-Betanzos, Manuel Octavio Ramírez-Sucre, Juan Valerio Cauich-Rodríguez, Ingrid Mayanin Rodríguez-Buenfil

**Affiliations:** 1Center for Research and Assistance in Technology and Design of the State of Jalisco, A.C Southeast Sub-Headquarters, Cadastral Record 31264, Km 5.5 Sierra Papacal–Chuburna Puerto Highway, Scientific and Technological Park of Yucatan, Mérida 97070, Mexico; jofernandez_al@ciatej.edu.mx (J.F.-C.); keaviles_al@ciatej.edu.mx (K.A.A.-B.); oramirez@ciatej.mx (M.O.R.-S.); 2Center for Scientific Research of Yucatán, Materials Unit, 43rd Street No. 130 × 32 and 34, Chuburna de Hidalgo, Mérida 97205, Mexico

**Keywords:** *Citrus aurantium*, by-products, natural deep eutectic solvents, phenolic profile

## Abstract

In the Yucatán Peninsula, *Citrus aurantium* L. has a strong cultural and culinary relevance where local industries already process its juice and essential oils, producing large amounts of by-products. In this context, green chemistry strategies have accelerated the valorization of agro-industrial residues, where Natural Deep Eutectic Solvents (NADESs) stand out due to their low cost, ease of preparation, and high extraction efficiency. This study focuses on evaluating different NADES combinations for the extraction of bioactive compounds from *C. aurantium* by-products, obtained after essential oil (cold pressing) and juice (mechanical pressing) extraction. A 3 × 2 × 2 factorial design was implemented to evaluate the effect of hydrogen bond donor (HBD: fructose, glucose and glycerol), molar ratio (MR: 1:1 and 1:2 mol/mol choline chloride (ChCl:HBD)) and added water (AW: 50 and 70%) on the polyphenolic profile, total phenolic content, total flavonoid content, ascorbic acid content and antioxidant capacity. HBD was the most critical factor in the extraction of bioactive compounds; the extract obtained with glycerol and 70% AW exhibited the highest hesperidin content (2186.08 mg/100 g dry mass), while the same HBD with 50% AW exhibited the highest quercetin + luteolin extraction (721.32 mg/100 g dry mass), both at the same MR (1:1 mol/mol). Glycerol also achieved the highest recovery of total flavonoids (1829.7 ± 17.85 mg quercetin equivalent/100 g dry mass) with an MR of 1:2 mol/mol and 70% AW. Finally, all other maximum values were obtained with fructose-based NADESs: the highest total phenolic content (3603. 7 ± 52.9 mg gallic acid equivalent/100 g dry mass) was achieved at an MR of 1:1 mol/mol and 50% AW, while for both vitamin C (1964.8 ± 33.7 mg ascorbic acid equivalent/100 g dry mass) and antioxidant capacity (84.31% inhibition), the maximum was reached at an MR of 1:2 mol/mol and 50% AW.

## 1. Introduction

Worldwide, the agri-food industry generates more than 190 million tons of by-products annually. This biomass, commonly referred to as waste, constitutes a valuable source of compounds such as sugars, dietary fiber, lipids, proteins, minerals, and bioactive molecules including phenolic acids, tannins, flavonoids and carotenoids. However, it is frequently discarded in landfills or other waste disposal sites, where its slow and progressive degradation contributes to the release of greenhouse gases, air, water, and soil contamination, and the occupation of vast areas that could be used for agriculture. The above highlights the importance of developing strategies for their appropriate disposal, recycling, or valorization. Among the most common by-products are those derived from the harvesting of oilseeds, vegetables, and horticultural crops, as well as residues obtained from plants such as grapes, pomegranates, melons, and particularly citrus fruits [1,2,3].

Citrus fruits are recognized for their elevated content of vitamin C and carotenoids with pro-vitamin A activity, particularly β-cryptoxanthin [4], positioning them as one of the most valued fruit categories due to their substantial nutritional and pharmacological relevance. Their distinctive aroma and flavor further contribute to their global importance, making them one of the most extensively cultivated crops, with an annual production exceeding 100 million tons, concentrated primarily in six countries: China (46.6 Mt), Brazil (18.8 Mt), India (14.3 Mt), Mexico (8.8 Mt), and the United States (6.2 Mt) [5,6,7]. Given the magnitude of this industry, citrus cultivation and processing generate considerable quantities of non-edible by-products, estimated to be 40 million tons of biomass per year, thereby underscoring the importance of developing efficient strategies for the valorization and sustainable management of these residues [8].

Citrus production is essential for the Mexican economy, contributing to 34.89% of the national agricultural Gross Domestic Product (GDP). In 2016, oranges represented the perennial crop with the largest cultivated area, covering 335,336 hectares. Citrus fruits are not only widely consumed domestically but also contribute significantly to exports, positioning Mexico as one of the leading countries in the global citrus market, particularly for sweet oranges (*Citrus sinensis* L. Osbeck), limes (*C. aurantifolia*, *Swingle* and *C*. *latifolia Tanaka*), and lemons (*C. limon* L.) [5,9]. In addition, other species, although cultivated on a smaller scale, have important cultural and culinary relevance in Mexican gastronomy, as in the case of *Citrus aurantium*.

Commonly known as sour orange or bitter orange, *C. aurantium* belongs to the Ru-taceae family, and its leaves, flowers, and fruits have been widely used in traditional Chinese medicine [10]. In Mexico, sour orange is primarily cultivated in the Yucatán Peninsula, where it plays a central role in the preparation of traditional dishes such as pibikek’en (“Cochinita pibil”, Yucatan-style slow-roasted pork) and xréenteja jetel k’ek’en (“Frijol con Puerco”, pork and beans) [11]. Although its cultivation is largely confined to backyard orchards, vulnerable communities in the region often grow it as livelihood, selling the fruit to local companies for the extraction of juice and essential oils [12]. This practice generates substantial amounts of residues, mainly consisting of albedo, flavedo, pulp, and seeds, which are commonly disposed of into the environment for natural degradation or alternatively sold as forage. In either case, this biomass is frequently undervalued, despite representing a rich source of bioactive compounds that could be harnessed for value-added applications.

Among the wide variety of bioactive compounds present in citrus by-products, polyphenols are currently among the most valued and extensively studied due to their broad spectrum of therapeutic effects against various diseases and their ability to prevent the onset of neurodegenerative disorders such Alzheimer and Parkinson’s disease [13]. However, their extraction is not a straightforward task, as these compounds are highly sensitive to factors such as pH, light, and heat, which can promote degradation, and consequently, reduce their functionality. Therefore, the selection of an appropriate extraction method becomes a crucial step [14]. In this context, green chemistry has emerged as an environmentally friendly alternative to conventional extraction techniques where there are disadvantages, such as the use of polluting solvents, energy consumption, long extraction times, toxicity and/or degradation of thermo-sensitive compounds, and offer advantages such as recognition as Generally Recognized as Safe (GRAS), shorter extraction times, biodegradability, and reduced costs [15,16].

Natural Deep Eutectic Solvents (NADESs) are considered a GRAS extraction system, as they are composed of metabolites naturally produced by plants, such as sugars, organic acids, and amino acids. They consist of mixtures of two or three components that do behave as hydrogen bond donors (HBDs) and hydrogen bond acceptors (HBAs) interacting through hydrogen bonding and resulting in a melting point lower than that of each individual component when combined in the appropriate molar ratio [17]. Due to the wide range of possible combinations, NADESs exhibit selective extraction capacity; in other words, depending on their composition and molar proportion, a given NADES can be adjusted to efficiently extract specific compounds [18]. Furthermore, since they are non-toxic and safe for human consumption, NADESs represent a highly promising alternative for the extraction of bioactive compounds from agro-industrial by-products [19].

To the best of our knowledge, the use of NADESs for the extraction of bioactive compounds from by-products of *C. aurantium* harvested under the distinctive conditions of the Yucatán Peninsula, mainly Leptosol soils [20] and warm sub-humid climate [21], after the extraction of juices and essential oils has not yet been reported. Therefore, the aim of this study was to evaluate different NADES combinations on the polyphenolic profile of *C. aurantium* by-products, with a view to their potential application in food, pharmaceutical, or cosmetic formulations.

## 2. Results

### 2.1. Evaluation of Total Polyphenols, Flavonoids, Ascorbic Acid, and Antioxidant Capacity in C. aurantium By-Products Extracts

Total polyphenol content (TPC; expressed as GAE content), total flavonoid content (TFC; expressed as QE content), total ascorbic acid (TAA; expressed as ascorbic acid equivalent) and antioxidant capacity (Ax; expressed as %DPPH inhibition) results are summarized in Figure 1; also, all corresponding numerical data are provided in the Appendix A. Additionally, the chromatogram of the extract obtained using NADESs formulated with glycerol as HBD, 1:1 MR and 70% AW, as is shown in Appendix A.

Significant differences (*p* < 0.05) in TPC (Figure 1a) were found among the experiments. The highest concentration of TPC was 3603.7 ± 52.9 mg GAE/100 g DM and it was achieved with NADESs based on fructose (MR 1:1 mol/mol and AW 50%). In contrast, the lowest concentrations: 1924.9 ± 69, 1955.2 ± 29.9, 1973.1 ± 53.5 and 1993.4 ± 16.9 mg GAE/100 g DM were obtained with glycerol- and glucose-based NADESs at experiments #11 (HBD glycerol; MR 1:2 mol/mol; AW 70%), #3 (HBD glucose; MR 1:1 mol/mol; AW 50%), #8 (HBD glycerol; MR 1:1 mol/mol; AW 70%) and #9 (HBD glucose; MR 1:1 mol/mol; AW 70%), respectively.

When comparing the effect of the HBD, fructose-based NADESs showed consistently higher TPC values (2643.3–3603.7 mg GAE/100 g DM) than glycerol (1924.9–2088 mg GAE/100 g DM) and glucose (1955.2–2128.8 mg GAE/100 g DM). Regarding the AW, most treatments exhibited higher values at 50% than at 70%. For example, Exp #1 (HBD fructose; MR 1:1 mol/mol; AW 50%) decreased from 3603.7 ± 52.9 to 2643.3 ± 33.8 mg GAE/100 g DM at 70% AW (Exp #7), and Exp #2 (HBD glycerol; MR 1:1 mol/mol; AW 50%) decreased from 2062.7 ± 84 to 1973.1 ± 53.5 mg GAE/100 g DM at 70% AW (Exp #8). A similar trend was observed for glucose-based NADESs at Exp #6 (HBD glucose; MR 1:2 mol/mol; AW 50%) with 2128.8 ± 44.3 mg GAE/100 g DM vs. Exp #12 (HBD glucose; MR 1:2 mol/mol; AW 70%) with 2124.8 ± 34.7 mg GAE/100 g DM (*p* > 0.05).

The total flavonoid content (TFC) was also determined in each experiment, and the results are summarized in Figure 1b. The highest flavonoid concentration was 1829.8 ± 17.8 mg QE/100 g DM, and it was obtained in experiment #11 (HBD glycerol; MR 1:2 mol/mol; AW 70%), while Exp #1 (HBD fructose; MR 1:1 mol/mol; AW 50%), Exp #3 (HBD glucose; MR 1:1 mol/mol; AW 50%), Exp #9 (HBD glucose; MR 1:1 mol/mol; AW 70%), and Exp #10 (HBD fructose; MR 1:2 mol/mol; AW 70%) yielded the lowest values (1161.4 ± 17.5, 1159.6 ± 73.6, 1221.1 ± 17.7, and 1212.8 ± 46.2 mg QE/100 g DM, respectively), with no significant statistical differences among them (*p* > 0.05). Extracts obtained from glycerol-based NADESs showed the highest flavonoid content (1363.4–1829.8 mg QE/100 g DM), compared to those prepared with fructose (1161.4–1385 mg QE/100 g DM) or glucose (1159.6–1443.8 mg QE/100 g DM).

In fructose-based NADESs, when the added water was 50%, flavonoid levels significantly increased as the molar ratio shifted from 1:1 mol/mol (Exp #1; 1161.4 ± 17.5 mg QE/100 g DM) to 1:2 mol/mol (Exp #4; 1385 ± 52.9 mg QE/100 g DM). A comparable trend was found in glucose-based NADESs, where with 50% added water, flavonoid content rose from 1159.6 ± 73.6 mg QE/100 g DM at MR 1:1 mol/mol (Exp #3) to 1313.9 ± 97.8 mg QE/100 g DM at MR 1:2 mol/mol (Exp #6). In contrast, for glycerol-based NADESs with 50% water, the opposite effect occurred: increasing the MR from 1:1 mol/mol (Exp #2) to 1:2 mol/mol (Exp #5) slightly reduced flavonoid content (1607.9 ± 46 to 1555.3 ± 46 mg QE/100 g DM) (*p* > 0.05). Overall, except for fructose-based NADESs, each hydrogen bond donor reached its highest flavonoid extraction capacity under conditions of MR 1:2 mol/mol and AW 70% (1443.8 ± 31.2 mg QE/100 g DM for glucose and 1829.7 ± 17.8 mg QE/100 g DM for glycerol).

Total ascorbic acid (TAA) of the extracts varied depending on the HBD, MR, and AW, with values ranging from 1635.3 ± 52.8 to 1964.9 ± 33.7 mg/100 g DM (Figure 1c). The highest concentrations were obtained using fructose as HBD in Exp #4 (MR 1:2 mol/mol and AW 50%) with 1964.9 ± 33.7 mg/100 g DM and Exp #1 (MR 1:1 mol/mol and AW 50%) with 1948.2 ± 38 mg/100 g DM, while the lowest was observed with glycerol-based NADESs in Exp #5 (MR 1:2 mol/mol and AW 50%), reaching 1635.2 ± 52.8 mg/100 g DM. In general, extracts obtained with fructose yielded higher ascorbic acid values (1828.5–1964.8 mg/100 g DM) compared to those obtained with glucose (1719.5–1854.9 mg/100 g DM) and glycerol (1635.3–1892.5 mg/100 g DM). For fructose- and glucose-based NADESs, increasing the molar ratio from 1:1 to 1:2 mol/mol at 50% AW slightly enhanced ascorbic acid recovery, whereas in glycerol-based NADESs, this change did not result in an improvement. Regarding the effect of water added, decreasing AW to 50% increased ascorbic acid content in fructose-based NADESs, whereas in glycerol-based NADESs, the opposite trend was observed, with lower TAA values at lower water content.

DPPH Radical Scavenging assay was used to determine the antioxidant capacity of extracts (Ax). Ax of the extracts ranged from 79.22 ± 1.36% to 84.31 ± 1.15% among different experiments (*p* < 0.05) (Figure 1d). The highest Ax was obtained with fructose-based NADESs at Exp #4 (MR 1:2 mol/mol and AW 50%), exhibiting 84.31 ± 1.15% inhibition, while the lowest were observed with Exp #12 (HBD glucose; MR 1:2 mol/mol; AW 70%) with 79.22 ± 1.36% inhibition, Exp #8 (HBD glycerol; MR 1:1 mol/mol; 70% AW) with 79.99 ± 0.17% inhibition, Exp #6 (HBD glucose; MR 1:2 mol/mol; 50% AW) with 80.03 ± 0.14% inhibition, Exp #5 (HBD glycerol; MR 1:2 mol/mol and AW 50%) with 80.27 ± 0.25% inhibition, Exp #11 (HBD glycerol; MR = 1:2 mol/mol; AW 70%) with 80.31 ± 1.36% inhibition and Exp #2 (HBD glycerol; MR 1:1 mol/mol; AW 50%) with 80.52 ± 0.29% inhibition. Overall, fructose-based NADESs exhibited the strongest antioxidant capacity (82.26–84.31% inhibition), followed by glycerol (79.99–80.52% inhibition) and glucose (79.22–81.71% inhibition). At 50% AW, increasing the MR from 1:1 to 1:2 mol/mol slightly reduced antioxidant activity in glucose-based NADESs from 81.57 (Exp #3) to 80.03% inhibition (Exp #6) (*p* < 0.05) and glycerol-based NADESs from 80.52 (Exp #2) to 80.27% inhibition (Exp #5) (*p* > 0.05). In contrast, fructose-based NADESs exhibited the opposite effect, with antioxidant activity increasing from 82.26 (Exp #1) to 84.31% inhibition (Exp #4) (*p* < 0.05).

Figure 2 demonstrates the significant effect of all factors and interactions on different responses. HBD was the main factor influencing all response variables. For TPC (Figure 2a), HBD (A) presented the largest standardized effect, followed by AW (C) and their interaction (AC).

In TFC (Figure 2b), significant effects were observed for HBD and MR (B). However, a triple interaction among three main factors was observed (ABC). For TAA (Figure 2c), HBD was the dominant factor, with additional contributions from AC and AB interactions. Finally, in Ax (Figure 2d), HBD was the only factor with a clear effect.

### 2.2. Evaluation of the Profile of Phenolic Compounds in C. aurantium By-Products Extracts

Among the 16 polyphenols examined in each extract from the factorial design (Appendix A), only 13 individual polyphenols were identified and quantified. Gallic acid, apigenin and diosmetin were not detected in any of the experiments (Appendix A). Flavonoids (Figure 3) and phenolic acids (Figure 4) contents are displayed as bar charts to enhance clarity and facilitate comparison among treatments.

The extraction profiles of the main flavonoids in *C. aurantium* by-products showed clear differences depending on the type of NADES and the extraction conditions. Overall, hesperidin was consistently the predominant compound across all treatments, reaching its maximum concentration when glycerol-based NADESs were used at a 1:1 mol/mol MR and 70% AW (Exp #8), obtaining 2186 ± 2.06 mg/100 g DM.

In general, glycerol-based NADESs provided the highest hesperidin recovery (1386.74–2186.08 mg/100 g DM). Conversely, neohesperidin was the least abundant flavonoid among the main ones, where a glucose-based NADESs showed the greatest efficiency extracting this compound, reaching 141.99 ± 2.62 mg/100 g DM under experiment #6 (HBD glucose; MR = 1:2 mol/mol; AW 50%).

Regarding the remaining flavonoids, quercetin + luteolin represented the second main flavonoids, with concentrations ranging from 229.69 to 721.32 mg/100 g DM across treatments. The highest yields (721.32 ± 23.05 and 707.20 ± 13.17 mg/100 g DM) were observed in Exp #2 (HBD glycerol; MR 1:1 mol/mol; AW 50%) and Exp #10 (HBD fructose; MR 1:2 mol/mol; AW 70%), respectively (*p* > 0.05). Kaempferol and rutin were also present in all treatments. Kaempferol showed its highest recoveries in Exp #9 (HBD glucose; MR 1:1 mol/mol; AW 70%), Exp #5 (HBD glycerol; MR 1:2 mol/mol; AW 50%), Exp #8 (HBD glycerol; MR 1:1 mol/mol; AW 70%), and Exp #11 (HBD glycerol; MR 1:2 mol/mol; AW 70%), with values of 256.65 ± 3.30, 256.79 ± 0.77, 257.08 ± 3.03, and 266.88 ± 8.44 mg/100 g DM, respectively (*p* > 0.05). Meanwhile, rutin reached its maximum concentration (121.16 ± 9.10 mg/100 g DM) under experiment #11 (HBD glycerol; MR 1:2 mol/mol; AW 70%). Overall, NADESs formulated with glycerol as HBD stood out as the most effective for extracting the different flavonoids identified.

The extraction of phenolic acids from *C. aurantium* by-products also varied significantly depending on the type of NADES and extraction conditions. Chlorogenic acid was mainly recovered in treatments with fructose- and glucose-based NADESs, reaching the highest concentration in Exp #4 (HBD fructose; MR 1:2 mol/mol; AW 50%), Exp #10 (HBD fructose; MR 1:2 mol/mol; AW 70%), Exp #3 (HBD glucose; MR 1:1 mol/mol; AW 50%) and Exp #2 (HBD glycerol; MR 1:1 mol/mol; AW 50%) with 44.01 ± 0.05, 44.28 ± 0.44, 44.14 ± 1.05 and 43.60 ± 1.10 mg/100 g DM, respectively (*p* > 0.05). Cinnamic acid displayed a similar trend; even though its maximum concentration reached 27.59 ± 0.83 mg/100 g DM, there was no statistical difference among 10 out of 12 treatments. Coumaric acid was consistently found at lower concentrations across treatments, with the highest extraction obtained at Exp #7 (13.86 ± 0.08 mg/100 g DM) and Exp #8 (13.93 ± 0.08 mg/100 g DM), both at a 1:1 MR mol/mol and 70% of AW using fructose and glycerol, respectively, as HBDs.

Regarding vanillin, its extraction was limited to specific treatments. The highest concentration was obtained with fructose-based NADESs, specifically Exp #4 (MR 1:2 mol/mol and AW 50%), Exp #7 (MR 1:1 and AW 70%), and Exp #10 (MR 1:2 and AW 70%), with values of 9.02 ± 0.10, 9.32 ± 0.01 and 9.07 ± 0.06 mg/100 g DM, respectively (*p* > 0.05). In general, fructose and glucose as HBDs favored the extraction of chlorogenic acid and vanillin, whereas glycerol-based NADESs were less efficient for these specific compounds but still enabled the recovery of cinnamic acid at comparable levels.

Other minor flavonoids and phenolic acids were also found in the extracts, but their extraction was very selective (Appendix A); for example, protocatechuic acid and catechin were only found in fructose-based NADESs, both reaching their maximum at experiment #10 (MR 1:2 mol/mol and AW 70%) obtaining 62.82 ± 7.6 and 23.09 ± 5.6 mg/100 g DM, respectively. Naringenin, another characteristic flavonoid in citrus species, was also found in sugars-based NADESs, reaching its highest concentration at Exp #4 (HBD fructose; MR 1:2 mol/mol; AW 50%) and Exp #3 (HBD glucose; MR 1:1 mol/mol; AW 50%) reaching 25.59 ± 1.10 and 25.07 ± 0.05 mg/100 g DM, respectively (*p* > 0.05).

Table 1 presents the *p*-values of the main factors and their interactions. HBDs (A) showed a significant effect on all the individual polyphenols evaluated. In addition, the molar ratio (B) and added water (C) also influenced the extraction of most compounds, except for chlorogenic acid (only B showed no effect), rutin (only C showed no effect), cinnamic acid, hesperidin, vanillin, and naringenin. Regarding interactions, although double interactions (AB, AC, and BC) were observed, a triple interaction (ABC) was particularly evident for most individual polyphenols, except for chlorogenic acid, coumaric acid, and quercetin + luteolin.

### 2.3. Pearson Correlation

The Pearson correlation coefficient is a statistical measure that quantifies the strength and direction of relationship between two quantitative variables. Figure 5 presents the heatmap of Pearson correlations obtained from all response variables in the 3 × 2 × 2 experimental design.

A positive correlation was observed between TPC and TAA (r = 0.7120), as well as between TPC and Ax (r = 0.7275). Additionally, some individual polyphenols displayed strong associations, such as hesperidin with cinnamic acid (r = 0.9108) and vanillin with coumaric acid (r = 0.7559). In contrast, certain secondary metabolites exhibited negative correlations; for instance, naringenin with kaempferol (r = −0.8021) and naringenin with cinnamic acid (r = −0.7211). Overall, most correlations showed no statistical relevance, with coefficients below 0.7.

### 2.4. Principal Component Analysis

Principal component analysis (PCA) with K-means clustering was performed on the experimental conditions of the 3 × 2 × 2 factorial design (Figure 6). PCA shows that the two principal components accounted for 59.23% of the total variance. These components are presented as a biplot with K-means clustering, simultaneously displaying the scores, loadings, and cluster groupings of the first two principal components. However, the three main components accounted for 73% of the total variance and complementary biplots of PC1 vs. PC3 and PC2 vs. PC3 are provided in Appendix A, allowing further visualization of patterns across all three principal components.

**Table 2 molecules-30-04551-t002:** Experimental design 3 × 2 × 2 for obtaining an extract rich in bioactive compounds from industrial by-products of *C. aurantium* using different NADESs.

#EXP	Encoded Values			Real Values			Response Variables *
	X_1_	X_2_	X_3_	HBD	MR (mol/mol)	AW (%)	
1	−1	−1	−1	Fructose	1	50	Y_1_
2	0	−1	−1	Glycerol	1	50	Y_2_
3	1	−1	−1	Glucose	1	50	Y_3_
4	−1	1	−1	Fructose	2	50	Y_4_
5	0	1	−1	Glycerol	2	50	Y_5_
6	1	1	−1	Glucose	2	50	Y_6_
7	−1	−1	1	Fructose	1	70	Y_7_
8	0	−1	1	Glycerol	1	70	Y_8_
9	1	−1	1	Glucose	1	70	Y_9_
10	−1	1	1	Fructose	2	70	Y_10_
11	0	1	1	Glycerol	2	70	Y_11_
12	1	1	1	Glucose	2	70	Y_12_

Note: #EXP = experiment number; HBD = hydrogen bond donor; MR = molar ratio of HBD (fructose, glycerol and glucose) per 1 mol of choline chloride; AW = percentage of added water to NADES; * the response variables (Y_n_) for every experiment are total polyphenol content (TPC), total flavonoid content (TFC), total ascorbic acid (TAA) and antioxidant capacity (Ax).

In this representation, the points correspond to the observations (experimental conditions), illustrating their distribution and grouping in the multivariate space, while the vectors represent the response variables (TPC, TFC, TAA, Ax, and the polyphenols measured, excluding gallic acid, protocatechuic acid, catechin, apigenin, and diosmetin), indicating both their contribution to the components and their correlations with each other. The superimposed clusters highlight groups of experimental conditions and variables that share similar profiles, facilitating the interpretation of how specific variables drive the separation or clustering within the dataset.

As observed, the distance between the eigenvectors of TAA, TPC, and Ax is very small, indicating that samples with high TPC values also contain considerable levels of TAA and Ax. In contrast, TFC is positioned nearly opposite, suggesting that the first three variables and TFC are inversely related. Regarding the individual polyphenols, all acids except cinnamic acid are located on the right side of the PCA, while all flavonoids except naringenin are positioned on the left.

Finally, four distinct clusters of experimental conditions were observed. The first cluster (green ellipse) groups extractions 2, 3, and 5, which are associated with high levels of naringenin and vanillin. The second cluster (yellow ellipse) includes extractions 4, 6, 7, and 10, characterized by higher TPC, TAA, and Ax values, indicating a strong positive correlation among these variables. The third cluster (red ellipse) comprises extractions 11 and 12, which are associated with rutin, quercetin, and luteolin, while the fourth small cluster (purple ellipse) groups extractions 8 and 9, which are presented as intermediate polyphenol profiles.

## 3. Discussion

The composition of NADESs proved to be a determining factor in the extraction of phenolic compounds from *C. aurantium*. The results for TPC, TFC, TAA, and Ax showed significant differences among treatments. In all cases, choline chloride was used as the common HBA, ensuring that the observed variations were solely due to the different HBDs, MR, and AW employed. The Folin–Ciocalteu method, widely used for quantifying polyphenols, is based on an electron transfer reaction in which the antioxidant species acts as an electron donor and the reagent serves as an oxidizing agent [22].

Experiment #1 (HBD fructose; MR 1:1; AW 50%) achieved the highest TPC (3603.7 ± 52.9 mg GAE/100 g DM), which is higher than the value reported by Edrisi et al. [23], who extracted 785 mg GAE/100 g DM using a NADES composed of choline chloride as the HBA and 1,4-butanediol as the HBD after an optimization process (optimal conditions: temperature of 40 °C, extraction time of 30 min, water content of 49.95%, and a solvent-to-biomass ratio of 18.97:1). These differences could be attributed to variations in the raw material and sample preparation methods. While Edrisi et al. [23] used only the peel, the raw material in our study consisted of a mixture of different citrus fruit parts (albedo, flavedo, pulp, and seeds). In addition, the moisture removal method employed by those authors was air-drying at 25 °C, whereas in the present study, lyophilization was used. This technique can better preserve sensitive bioactive compounds due to the low temperatures and absence of light exposure [24]. These results are comparable to those obtained using organic solvents. For instance, Abdallah et al. [25] who optimized a solid-to-liquid ratio of 2.4 g/100 mL, an acetone concentration of 80%, and a sonication time of 34.7 min, achieved an extraction yield of 3376 ± 563 mg GAE/100 g DM. However, it is important to highlight that extracts obtained with organic solvents require purification steps to remove toxic residues, whereas NADES-based extracts, being made of natural plant metabolites, are considered safe and classified as GRAS [16]. In contrast, the values obtained in the present study are lower than those reported by Amador-Luna et al. [26], who developed an innovative three-step sequential extraction process: (1) Supercritical Fluid Extraction (SFE) for the nonpolar fraction, (2) Pressurized Liquid Extraction (PLE) using NADESs for polar compounds, and (3) an antisolvent extraction step to recover compounds adsorbed in the PLE residue. The NADES extract, formulated with choline chloride and glycerol at a 1:2 mol/mol MR with 57.9% AW, yielded 6240 mg GAE/100 g DM at semi-pilot scales. Although glycerol was also used in this study as HBD in a 1:2 mol/mol MR, the lower phenolic yields may be attributed to two main factors. First, the NADES used by Amador-Luna et al. [26] were pressurized, which involves operating at high temperature and pressure (below critical point), enhancing solubility and mass transfer within the matrix and allowing rapid and efficient extraction with minimal solvent use [27]. Second, the raw material used in the present study had already undergone essential oil removal (via cold pressing extraction), and as reported by Amador-Luna et al. [26] and other authors such as Alizadeh [28], essential oils themselves contain phenolic compounds (664 and 4135 mg GAE/100 g, respectively). However, the results obtained in this work confirm that even after essential oil extraction, *C. aurantium* industrial by-products still retain significant amounts of these bioactive phenolic compounds.

In the case of total flavonoids, experiment #11 (HBD glycerol; MR 1:2; AW 70%) showed the highest TFC (1829.7 ± 17.8 mg QE/100 g DM). This behavior, according to Chatterjee et al. [29], Panić et al. [30] and Tiecco et al. [31], is related to the intrinsic microheterogeneity of NADESs, defined as localized nanoscale variations in structure, and to the transition toward a “water in DES” regime at high water contents. Under these conditions, the original hydrogen-bond network of the NADES becomes partially disrupted and persists as dispersed microdomains within a predominantly aqueous continuous phase. This creates a more dynamic, less viscous, and highly microheterogeneous environment in which solute–water interactions become increasingly relevant while the remaining NADES nanodomains still offer locally structured hydrogen-bonding sites [29]. This could explain why experiment #11, along with #5 and # 8, exhibited limited selectivity for phenolic acids such as chlorogenic acid and vanillin (Figure 4), since at high water contents the NADES contains fewer OH-rich regions arising from densely hydrogen-bonded HBA–HBD interactions, reducing the stabilization of these small phenolic acids. On the other side, considering that most flavonoids are glycosylated, the glycone can strongly interact with water through extensive hydrogen bonding, whereas the aglycone may preferentially interact with the residual HBA–HBD clusters. The coexistence of these two solvation environments can facilitate the solubilization of bulky flavonoid glycosides such as hesperidin. Notably, the microheterogeneous nature of choline chloride–glycerol NADESs has been previously suggested by Panić et al. [30], who reported that a choline chloride–glycerol NADES at 30% AW could extract both phenolic compounds and hydrophobic molecules such as D-limonene from orange peel waste, attributing this effect to the presence of microheterogeneity within the NADES structure.

These results are higher than those reported by Estrada-Sierra et al. [32], who obtained a TFC concentration of 830 mg QE/100 g DM from the by-products (seeds, pulp, and peel) of *C. aurantium* after juice extraction at the green maturity stage, using water as solvent for the extraction, this difference being attributed to the solvent used. Although water is a polar compound, flavonoids such as hesperidin are poorly water-soluble [33]. Conversely, NADESs, being more complex mixtures, can extract a wider diversity of molecules [17]. Also, the results obtained from by-products in this work are considerably lower than those reported by Liu et al. [34], who extracted 16,816 ± 47 mg QE/100 g of total flavonoids from *Aurantii Fructus Immaturus* (AFI), the dried immature fruit of *C. aurantium*. This high yield was achieved using ultrasound-assisted extraction with Natural Deep Eutectic Solvents (UAE-DESs). Specifically, the optimal solvent selected was a betaine–urea (Bet:Ur) mixture, and the extraction conditions were optimized through Response Surface Methodology (RSM). The optimal parameters included a molar ratio of 1:3, a water content of 30%, a liquid-to-solid ratio of 30:1 mL/g, an extraction temperature of 50 °C, and an extraction time of 50 min.

The differences observed may be attributed to the use of betaine as the hydrogen bond acceptor (HBA), which has an amphiphilic and zwitterionic nature, is modulated by pH, and can form micelle-like self-assemblies that act as nanocontainers with dual affinity for both hydrophilic and lipophilic compounds [35]. Furthermore, since the raw material used by Liu et al. [34] had not undergone essential oil removal, the extraction likely included less polar compounds associated with these fractions. According to Shraim et al. [36], such compounds can be quantified using the AlCl_3_, which is not an specific colorimetric assay; in other words, a lot of compounds can be quantified if they comply at least one of the following structural requirements: a carbonyl group at C-4 conjugated with the ring and a hydroxyl group at C-3 or C-5; a catechol system (3′,4′-OH) on ring B; or other accessible phenolic –OH groups, although the latter may form less stable complexes. Therefore, since betaine-based NADESs are highly effective at extracting diverse types of compounds, it is likely that the quantified phenolic content also includes non-flavonoid phenolic compounds.

Regarding to ascorbic acid content, to the best of our knowledge, its quantification in NADESs using spectrophotometry has not been previously reported. Fructose proved to be the most effective HBD, with experiments #1 (MR 1:1 mol/mol and AW 50%) and #4 (MR 1:2 mol/mol and AW 50%) showing the highest TAA levels with 1948.2 ± 38 and 1964.8 ± 33.7 mg/100 g DM, respectively. These values are considerably higher than those reported in the literature for any citrus species. For instance, in *C. aurantium*, values of 24 mg/100 g have been reported for immature fruit juice [37], while the peel of other citrus fruits such as lemon and grapefruit have shown maximum levels of 58.59 and 113.3 mg/100 g DM, respectively [38]. Such differences may be mainly attributed to the nature of the raw material used in this study, which consisted of a mixture of different parts of the fruit (albedo, flavedo, pulp, and seeds), potentially explaining the high ascorbic acid content. On the other hand, choline chloride–based NADESs with sugar (e.g., fructose and glucose as HBDs) have been reported to exhibit slightly acidic to near neutral pH values depending on their composition and water content [39]. This intrinsic acidity and high polarity may contribute to improved extraction efficiency and to the enhanced stability of ascorbic acid under the conditions applied in this study. Indeed, vitamin C (with no presence of oxygen) is known to be more stable under acidic environments (pH 5–6), which further supports the observed high ascorbic acid content in the extracts [40].

Antioxidant capacity was measured using the DPPH reagent, where antioxidants can act through two mechanisms: single electron transfer (SET), reducing the DPPH radical to its stable form, or hydrogen atom transfer (HAT), donating a H atom to the radical. In both cases, the radical is neutralized and the violet color fades to yellow [41]. Fructose proved to be the most effective HBD, with experiments #4 (MR 1:2 mol/mol and AW 50%), #7 (MR 1:1 mol/mol and AW 70%), and #10 (MR 1:2 mol/mol and AW 70%) showing the highest values among the experiments (*p* < 0.05), exhibiting inhibition percentages of 84.31% ± 1.15, 83.10% ± 1.56, and 82.85% ± 0.17, respectively (*p* > 0.05). These results are slightly lower than those reported by Ramírez-Sucre et al. [42], who achieved 100% DPPH radical inhibition in the peel of *C. aurantium*. This difference could be attributed to the pretreatment process of the raw material, since the peel was manually separated and did not undergo any industrial extraction process (essential oils and juice extraction), allowing polyphenolic compounds, which are highly sensitive to factors such as light and heat [14] to remain undegraded. The higher antioxidant capacity observed with fructose compared to glucose and glycerol could be attributed to its greater ability to extract phenolic acids. According to Gulcin et al. [43], the stability of the DPPH radical is largely due to steric hindrance around the divalent nitrogen atom and to the “push–pull” effect, which limits the approach of bulky molecules. Thus, large molecules such as glycosylated flavonoids encounter greater steric hindrance when donating electrons or hydrogen atoms due to their restricted access to the radical center. In contrast, smaller and less-hindered molecules, such as phenolic acids, can more easily approach and react rapidly, resulting in a higher percentage of DPPH radical inhibition.

Regarding the polyphenolic profile, hesperidin was the predominant flavonoid in the NADES extracts, reaching its highest concentration (*p* < 0.05) in experiment #8 (HBD: glycerol; MR: 1:1 mol/mol; AW: 70%) with 2186.08 ± 2.06 mg/100 g DM. These results are similar to those obtained by Liu et al. [34], who, under DES optimal extraction conditions (40% water in betaine/ethanediol (1:4) at 60 °C for 30 min heated extraction and a solid-to-liquid ratio of 1:100 g/mL), extracted 2872 ± 0.35 mg/100 g DM. However, it is important to note that under those conditions, they also extracted considerable amounts of naringin (3835 mg/100 g DM) and neohesperidin (8367 mg/100 g DM), the latter being much higher than the value obtained in the present study (141.99 mg/100 g DM). Glycerol proved to be the most effective HBD for flavonoid extraction, which, according to Qader et al. [44], can be explained by the structural nature of both molecules. Since hesperidin is glycosylated, it has a relatively large molecular volume that hinders the formation of hydrogen bonds with bulky and hydroxyl groups rich in HBDs such as sugars. In contrast, glycerol, being a smaller molecule with a reduced number of hydroxyl groups, offers greater conformational flexibility and lower steric hindrance, allowing it to interact more efficiently with the accessible sites of hesperidin.

A similar trend was observed for quercetin + luteolin, the second-most abundant (glycosylated) flavonoids across experiments, whose maximum concentration (721.32 ± 23.05 mg/100 g DM) was also obtained using glycerol as HBD (MR: 1:1 mol/mol; AW: 70%). These values are higher than those reported by Ramírez-Sucre et al. [42], who extracted 160.99 ± 1.75 mg/100 g from *C. aurantium* peel using a NADES composed of glucose as an HBD (MR: 1:2 mol/mol; AW: 70%).

Finally, in the case of phenolic acids, chlorogenic acid was reported in the highest amounts in experiments #4 (HBD fructose; MR 1:2 mol/mol; AW 50%), #10 (HBD fructose; MR 1:2 mol/mol; AW 70%), #3 (HBD glucose; MR 1:1 mol/mol; AW 50%) and #2 (HBD glycerol; MR 1:1 mol/mol; AW 50%) with 44.01 ± 0.05, 44.28 ± 0.44, 44.14 ± 1.05 and 43.60 ± 1.10 mg/100 g DM, respectively (*p* > 0.05). The three HBDs employed (glycerol, fructose, and glucose) showed similar extraction efficiencies (*p* > 0.05) for this compound. This lack of selectivity can be explained by the same reason that makes glycerol the most effective HBD for flavonoid extraction [44]. Due to their relatively small size and lower structural complexity, phenolic acids lack the steric hindrance that limits the solubilization of glycosylated flavonoids such as hesperidin. Their steric accessibility and the presence of hydroxyl groups in aromatic positions facilitate the formation of hydrogen-bond interactions with any of the tested HBDs. However, when analyzing each treatment individually, fructose with experiments #1 (MR 1:1 mol/mol and AW 50%) and #10 (MR 1:2 mol/mol and AW 70%) and glucose with experiments #3 (MR 1:1 mol/mol and AW 50%) and #12 (MR 1:2 mol/mol and AW 70%) enabled greater recovery in the main phenolic acids (chlorogenic acid, coumaric acid, cinnamic acid and vanillin), followed by glycerol in only experiment #2 (MR 1:1 mol/mol and AW 50%). This trend suggests that HBDs richer in –OH groups, such as sugars, may generate a denser network of interactions, favoring the solubilization of simple phenolic compounds, in contrast with glycerol, whose lower hydroxyl density makes it less competitive for this class of metabolites.

The wide variety of compounds extracted in this study represents a promising opportunity for their incorporation into food matrices. As previously mentioned, NADESs are solvents with very low toxicity, and are scientifically supported by studies such as those by Ferreira et al. [45] and Popović et al. [46], who evaluated toxicity and cytotoxicity parameters in in vivo and in vitro models, respectively, concluding that these extracts do not compromise cell viability or induce mortality. This major advantage allows the extracts to be directly added to food matrices, as demonstrated by Şen et al. [47], who incorporated NADES extracts into food products such as smoothies. The fortification of sunflower pomace-based NADES extracts enhanced the antioxidant efficiency of the final product, showing great potential as natural food additives. Furthermore, Avilés-Betanzos et al. [48], went a step further by microencapsulating NADES extracts to protect them from environmental factors such as light and heat, and subsequently incorporating them into an isotonic beverage. This study demonstrated the excellent bioaccessibility of the microencapsulated phenolic compounds in the digestive system under both postprandial and fasting conditions, as confirmed through in vitro digestion assays.

## 4. Materials and Methods

### 4.1. Plant Material

Fresh *C. aurantium* by-products were obtained from ARPEN Juicer S.P.R. de R.L. de C.V., located in Akil, Yucatán, Mexico (20°17′14.7″ N, 89°23′0.9″ W), immediately after the juice and essential oil extraction process. The raw material consisted of a mixture of albedo, flavedo, pulp, and seeds.

### 4.2. Bitter Orange Drying Leaf Process

The collected by-products were frozen at −20 °C for at least 48 h. Then, they were subjected to freeze-drying using a freeze-dryer (Feezone LABCONCO^®^, Kansas City, MO, USA) under pressure conditions of 0.005 mBar, temperature of −80 °C for 96 h, until they reached moisture content 3%. Dried *C. aurantium* by-products were ground using a grinder (Hukën×MasterChef^®^, Naucalpan de Juarez, EDOMEX, Mexico), the resulting powder was sieved (#35, 500 µm; Fisher Scientific, Boston, MA, USA). The flour obtained was stored in a desiccator, in darkness, and at room temperature (30 °C) until the extraction process.

### 4.3. Experimental Design

A 3 × 2 × 2 factorial experimental design (Table 2) was implemented to evaluate the effect of three independent variables on the extraction process: (1) HBD, tested at three levels: fructose, glucose, and glycerol; (2) molar ratio, evaluated at two levels (1:1 and 1:2 mol/mol) and (3) added water, tested at two levels (50% and 70%). This design generated a total of 12 experimental conditions, allowing the individual effects of each factor as well as their interactions to be efficiently analyzed.

### 4.4. NADES Preparation

The preparation of NADESs was performed using the heating and stirring method, as described by Mansinhos et al. [49]. The components (choline chloride as the hydrogen bond acceptor; molecular weight 139.63 g/mol) and fructose, glucose, and glycerol as hydrogen bond donors (molecular weights 180.15, 180.16, and 92.09 g/mol, respectively) were mixed at molar ratios of 1:1 and 1:2 (the donor ratio being the variable), at 90 °C until a transparent liquid was obtained (2 h). Subsequently, to reduce viscosity, water was added at 50% *w*/*w* and 70% *w*/*w* of the total mixture weight.

### 4.5. Ultrasound-Assisted Extraction of Polyphenols Using NADESs

Ultrasound-assisted extractions were performed according to Ramírez-Sucre et al. [42] with some modifications. For each extraction, 1 g of the sample was mixed with 12 mL of the corresponding NADES (According to experimental design) and vortexed. Subsequently, the samples were subjected to an ultrasonic bath (BRANSON^®,^ model 351, 42 kHz, Danbury, CT, USA) for 30 min and centrifuged at two steps: (a) at 4700 rpm for 30 min at 4 °C and (b) supernatant from step (a) was centrifuged at 15,000 rpm for 30 min at 4 °C (step b was performed 3 times, each time using the supernatant obtained from the previous centrifugation), with a tabletop centrifuge (Hettich^®^ Mikro 22R, Beverly, MA, USA). Extract samples were filtered with a nylon filter (0.22 μm) and placed in amber chromatographic vials under refrigeration (<18 °C) until analysis.

### 4.6. Spectrophotometric Assays

#### 4.6.1. Total Polyphenol Content Determination

Total polyphenol content (TPC) was determined following the method described by Singlenton et al. [50], with modifications. Initially, a dilution (1:50) of each extract was made with distilled water, and the determination of TPC was carried out by the spectrophotometric method using the Folin–Ciocalteu reagent. Briefly, 1 mL of each diluted sample was mixed with 3 mL of distilled water and 250 μL of Folin–Ciocalteu reagent diluted at a 1:2 *v*/*v* ratio. After 5 min, 750 μL of 20% Na_2_CO_3_ and 950 μL of distilled water were added. The resulting solutions were vortexed and incubated in the dark for 30 min. Finally, absorbance was measured at 765 nm using a UV–Vis spectrophotometer (Thermo scientific^®^, model Genesys 140, Mexico City, Mexico). For total polyphenols quantification, a calibration curve was made (Appendix A) using gallic acid from 5 µg/mL to 100 µg/mL (R^2^ = 0.9994), and TPC was expressed as mg of gallic acid equivalent (GAE)/100 g of dry mass (DM).

#### 4.6.2. Total Flavonoid Content Determination

Total flavonoid content (TFC) was determined following the method reported by Estrada-Sierra et al. [32]. Initially, a dilution (1:10) of each extract was made with distilled water, then, 100 μL of diluited sample was mixed with 250 μL of distilled water and 75 μL of 5% NaNO_2_. After 6 min incubation, 150 μL of 10% AlCl_3_ was added, and the mixture was left to stand for 5 min. Subsequently, 500 μL of NaOH (1 M) and 425 μL of distilled water were added. The resulting solutions were mixed and incubated in the dark for 30 min. Finally, absorbance was measured at 510 nm using a UV–Vis spectrophotometer (Thermo Scientific^®^ Genesys 140). Prior to the sample analysis, a calibration curve was generated (Appendix A) with quercetin from 100 µg/mL to 1400 µg/mL (R^2^ = 0.9993). The TFC of the samples was expressed as mg of quercetin equivalents (QE)/100 g of dry mass.

#### 4.6.3. Total Ascorbic Acid Determination

Total Ascorbic Acid content was determined according to Hagos et al. [51] with some modifications. Each solution to be evaluated was diluted 1:50 with an 80:20 water:acetonitrile mixture. Absorbance was measured at 244 nm using a Thermo Scientific^®^ Genesys 140 UV–Vis spectrophotometer. To quantify ascorbic acid, a five-point calibration curve was prepared (Appendix A) using standard ascorbic acid solutions at concentrations ranging from 5 to 50 μg/mL (R^2^ = 0.9997). Results were expressed as mg ascorbic acid equivalent/100 g of dry mass.

#### 4.6.4. Antioxidant Capacity Assessment

The antioxidant capacity (Ax) was assessed using the 2,2-diphenyl-1-picrylhydrazyl (DPPH) assay, following the method described by Chel-Guerrero et al. [52]. Briefly, 3.3 mg of DPPH was dissolved in methanol to a final volume of 100 mL, and the solution was adjusted to an absorbance of 0.700 ± 0.002 at 515 nm. After adjusting the DPPH solution, 100 µL of the sample was added to 3.9 mL of the adjusted DPPH solution. The mixture was vortexed and incubated in the dark for 30 min, after which absorbance (Abs) was recorded at 515 nm (UV–Vis spectrophotometer, Genesys 140). Antioxidant capacity was expressed as the percentage of inhibition, calculated according to Equation (1).$$\%\;\mathrm{DPPH}\;\mathrm{Inhibition}=\frac{\mathrm{Acontrol}-\mathrm{Asample}}{\mathrm{Acontrol}}\times \;100$$
where

Acontrol = absorbance of adjusted DPPH

Asample = absorbance of the DPPH solution with the sample

### 4.7. Polyphenol Profile

Polyphenol analysis and quantification in *C. aurantium* by-products were performed using an advanced Acquity UPLC H-Class system (Waters Corporation, Milford, MA, USA) equipped with a diode array detector (DAD)(Waters Corporation, Milford, MA, USA) and a high-performance column (Acquity UPLC HSS C18)(Waters Corporation, Wexford, Leinster, Irlanda) following the methodology adapted from Avilés-Betanzos et al. [53]. For accurate quantification, a calibration curve incorporating 16 distinct standards was employed, covering concentrations from 5 to 75 µg/mL. The standards were gallic acid, protocatechuic acid, chlorogenic acid, coumaric acid, cinnamic acid, catechin, rutin, kaempferol, quercetin + luteolin (quantified collectively, due to their overlapping peaks during analysis), vanillin, hesperidin, neohesperidin, naringenin, apigenin, and diosmetin. Analytical conditions were strictly controlled, with the column maintained at 45 °C and samples injected in 2 µL volumes. Detection was performed at 280 nm using a solvent system composed of water with 0.2% acetic acid for phase A and acetonitrile with 0.1% acetic acid for phase B. Each injection lasted 15 min, with the gradient elution program as follows: from 1% B to 30% B over 0 to 10 min, followed by a steady state of 30% B from 10 to 12 min, and finally returning to 1% phase B over 3 min.

### 4.8. Statical Analysis

Statistical analysis was performed using the software Statgraphics Centurion XVI version 16.1.03 (Statgraphics Technologies Inc.; Virgin, UT, USA) and R software version 4.0.3 (The R Foundation for Statistical Computing, Vienna, Austria). The studies were carried out randomly, measurements were taken in triplicate, and data were expressed as mean values ± standard deviation (SD). Pearson correlation analysis was performed to evaluate associations between individual polyphenols and spectrophotometric variables, while principal component analysis (PCA) was applied to reduce data dimensionality and visualize clustering patterns.

## 5. Conclusions

The composition of NADESs was found to be a key factor determining the extraction efficiency of bioactive compounds from *C. aurantium*. These results also align with the idea that the microstructural organization of NADESs, modulated by water content, can subtly influence solute selectivity. Among the HBDs tested, fructose and glycerol consistently showed superior performance, with fructose excelling in antioxidant capacity (DPPH inhibition) and total phenolic content, while glycerol was most effective for flavonoid extraction, particularly for glycosylated compounds such as hesperidin and quercetin + luteolin. Overall, these findings demonstrate that the rational selection of HBDs and NADES composition can be strategically used to target specific classes of bioactive compounds, enabling more efficient and selective extraction with potential applications in the development of functional ingredients and nutraceuticals from citrus by-products.

## Figures and Tables

**Figure 1 molecules-30-04551-f001:**
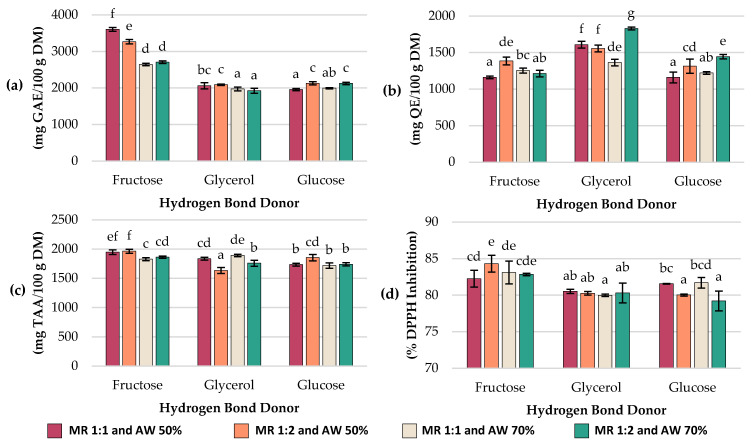
Characteristics of *C. aurantium* by-products NADES extracts: bar charts of (**a**) total polyphenol content (TPC), (**b**) total flavonoid content (TFC), (**c**) total ascorbic acid (TAA) and (**d**) antioxidant capacity (Ax). MR = molar ratio (mol/mol); AW = added water (%); GAE = gallic acid equivalent; QE = quercetin equivalent. Different superscript letters indicate statistically significant differences (LSD, *p* < 0.05). Values are means ± standard deviation (SD, n = 3).

**Figure 2 molecules-30-04551-f002:**
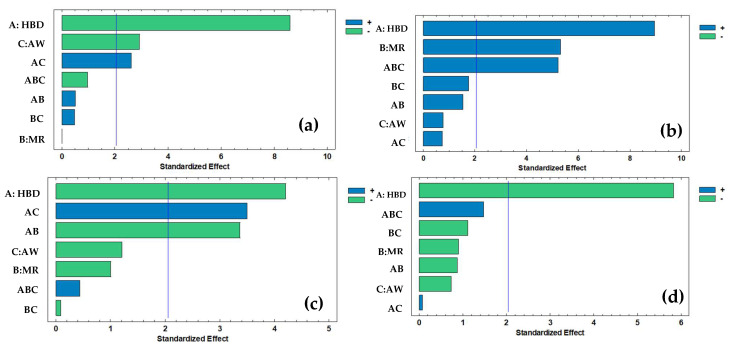
Pareto charts of (**a**) total polyphenol content (TPC), (**b**) total flavonoid content (TFC), (**c**) total ascorbic acid (TAA) and (**d**) antioxidant capacity (Ax). Where HBD = hydrogen bond donor; MR = molar ratio; AW = added water. Blue vertical line indicates standardized effect, bars beyond this line indicate a significant statistical effect (*p* < 0.05).

**Figure 3 molecules-30-04551-f003:**
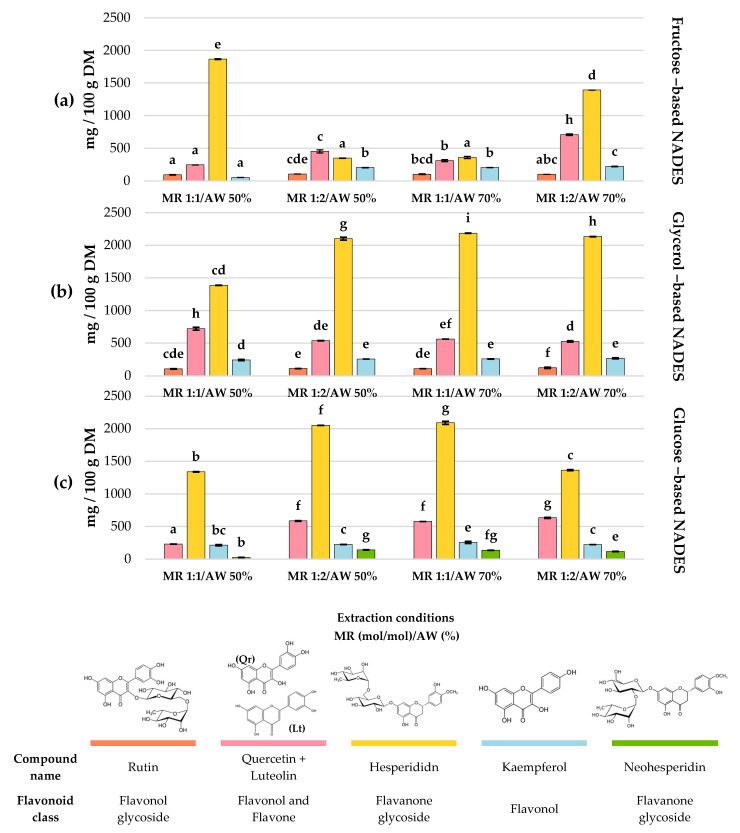
Major individual flavonoids in *C. aurantium* by-products extracts obtained using choline chloride-based NADESs with (**a**) fructose, (**b**) glycerol and (**c**) glucose as HBDs. Different letters on bars of the same color indicate statistically significant differences among individual flavonoids and HBDs (LSD, *p* < 0.05). MR = molar ratio (mol/mol); AW = added water (%); Molar ratio is expressed as 1 mol of HBA per mol of HBD.

**Figure 4 molecules-30-04551-f004:**
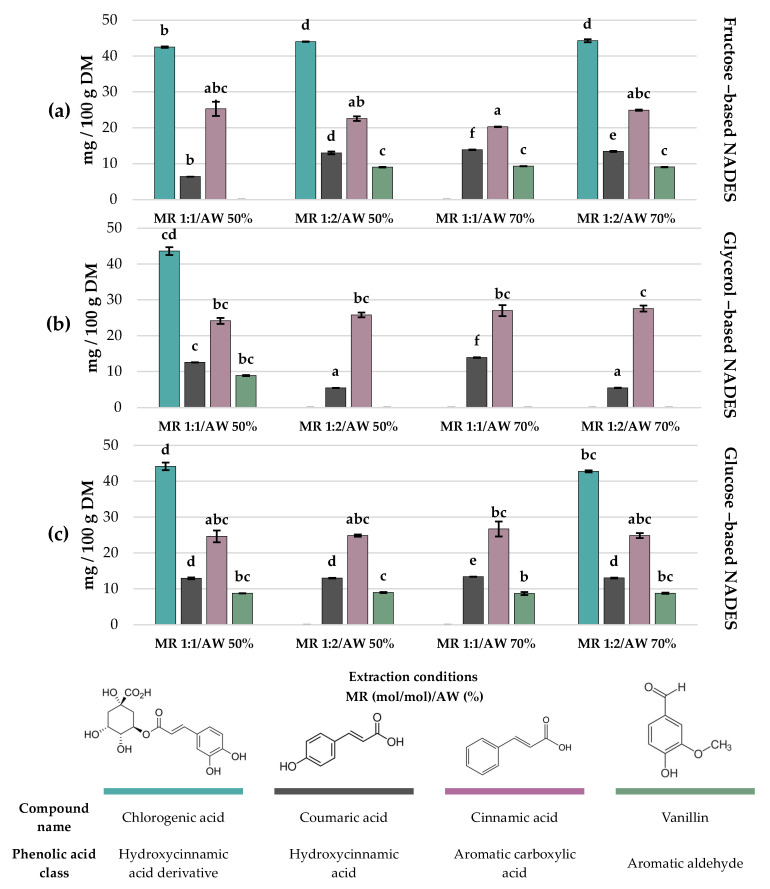
Major individual phenolic acids in *C. aurantium* by-product extracts obtained using choline chloride-based NADESs with (**a**) fructose, (**b**) glycerol and (**c**) glucose as HBDs. Different letters on bars of the same color indicate statistically significant differences among individual phenolic acids and HBDs (LSD, *p* < 0.05). MR = molar ratio (mol/mol); AW = added water (%). Molar ratio is expressed as 1 mol of HBA per mol of HBD.

**Figure 5 molecules-30-04551-f005:**
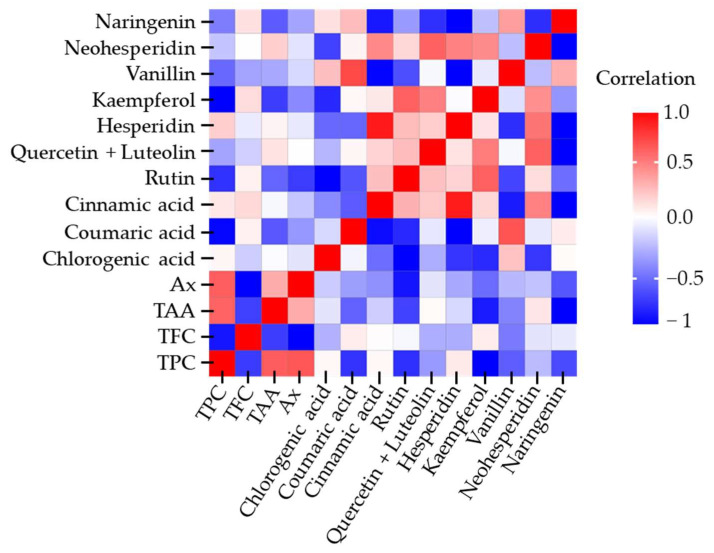
Pearson correlation heatmap of the 3 × 2 × 2 experimental design for the evaluation of polyphenol extraction from *C. aurantium* by-products using different NADESs. TPC = total polyphenol content; TFC = total flavonoid content; TAA = total ascorbic acid; Ax = antioxidant capacity.

**Figure 6 molecules-30-04551-f006:**
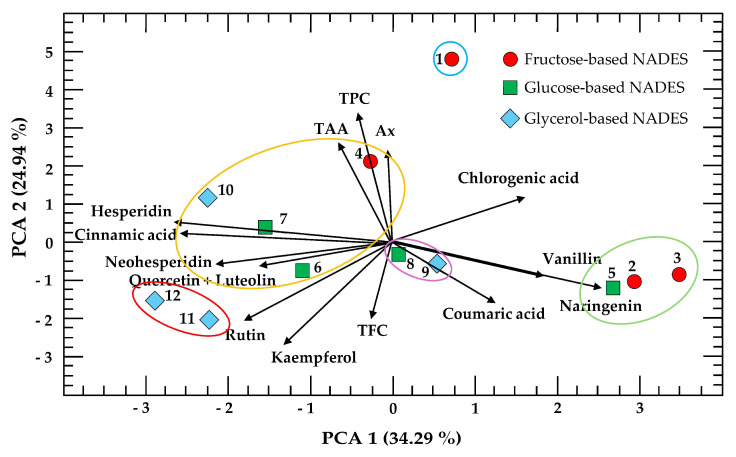
Principal component analysis (PCA) with K-means clustering, showing total polyphenol content (TPC), total flavonoid content (TFC), total ascorbic acid (TAA), antioxidant capacity (Ax), and the polyphenol profile. Numbers represent the different extraction conditions across the 12 experiments (Table 2). Numbers with the same shape indicate extractions that share the same HBD. Experiments grouped within clusters of the same color indicate similar characteristics.

**Table 1 molecules-30-04551-t001:** *p*-values of the main factors and interactions influencing the polyphenol profile in *C. aurantium* by-product extracts.

Individual Polyphenol	Main Factors and Interactions						
	A	B	C	AB	AC	BC	ABC
Protocatechuic acid	**0.0037**	**0.0137**	**0.0137**	**0.0037**	**0.0037**	**0.0137**	**0.0037**
Catechin	**0.0057**	**0.0192**	**0.0192**	**0.0057**	**0.0057**	**0.0192**	**0.0057**
Chlorogenic acid	**0.0001**	0.9715	**0.0009**	**0.0001**	0.9388	**<0.0001**	0.9629
Coumaric acid	**0.0238**	**0.0476**	**0.0368**	**<0.0001**	0.1007	0.0709	0.1149
Cinnamic acid	**0.0009**	0.4800	0.2571	0.9053	**0.0198**	0.2370	**0.0084**
Rutin	**0.0001**	**0.0052**	0.1023	0.4432	0.2213	0.1135	**0.0179**
Quercetin + Luteolin	**0.0025**	**0.0021**	**0.0229**	**0.0003**	**0.0138**	0.8641	0.8109
Hesperidin	**0.0001**	0.8547	0.6346	0.1349	0.0946	0.7030	**0.0003**
Kaempferol	**<0.0001**	**0.0060**	**0.0005**	**0.0058**	**0.0057**	**0.0037**	**0.0120**
Vanillin	**0.0018**	0.9833	0.9733	**0.0026**	**0.0020**	0.9282	**0.0021**
Neohesperidin	**0.0271**	**0.0106**	**0.0014**	0.0845	0.6498	0.0941	**0.0079**
Naringenin	**0.0124**	0.2887	0.2423	0.9439	0.9439	0.2690	**0.0124**

Note: A = hydrogen bond donor; B = molar ratio (mol/mol); C = added water (AW). Bold values indicate significant effects (*p* < 0.05) of main factors and/or their interactions on the concentration of individual polyphenols.

## Data Availability

The original contributions presented in this study are included in the article/Appendix A. Further inquiries can be directed to the corresponding authors.

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
