# Peer review of "Influence of Natural Deep Eutectic Solvent Compositions on the Polyphenol Profile of Citrus aurantium By-Products from Yucatán, México"

_molecules, 2025, doi:10.3390/molecules30234551_

Round 1
Reviewer 1 Report
Comments and Suggestions for Authors
1. Does the addition of water result in a ternary DES?
2. The characterization of DES is insufficient. For instance, the K-T values and other parameters have not been analyzed.
3. The performance data are comprehensive. Is the excellent performance related to the hydrogen-bonding microenvironment provided by the DES? How can this be verified?The surface electrostatic potential and density functional theory calculation method maybe is needed.
Reviewer 2 Report
Comments and Suggestions for Authors
This manuscript deals with the extraction, under ultrasound activation, of phenolic compounds from citrus aurantium co-products using a deep eutectic solvent of natural origin (NADES).
The authors attempted to determine, by UHPLC-DAD, the molecular profile of different extracts obtained with different NaDES, using polyphenolic and flavonoid standards as well as their glycosides. The aim of these extractions was to compare the different NaDES tested to understand the influence of each constituent of these solvents on hydrogen bonding and the influence of the presence of water. They also performed characterizations, including TPC, TFC, total ascorbic acid, and antioxidant capacity, particularly DPPH assays.
The work carried out is fairly standard and commonly encountered in the field of extracting antioxidant and/or polyphenolic molecules from plants. The manuscript is well-written but contains some errors and omissions that I would like the authors to correct:
- Proofread the manuscript, as there are many typographical errors and incorrect or missing words (l21, l651, l717, l722, …).
- It would be helpful to include a complete chromatogram describing the analysis of one of the extracts to show all the peaks and compounds present in the extracts, for which the authors used calibration standards and measured concentrations. This would demonstrate the effectiveness of the method for analyzing and separating the different constituents of the extracts and thus promote the method used.
- Table S1, presented in Supplementary Material, should be an integral part of the manuscript, and the choice of DOE parameters should be presented and included as experimental parameters. This would allow for a better understanding of which factors were targeted as variables.
- The reason for choosing the DPPH test is not clearly explained. Why not FRAP or CUPAC, for example? Does this relate to the intended application?
- Please check the bibliographic references, particularly their format, which is inconsistent throughout the reference section. For example, lines 814, 835, and 843. Ensure that the citation format conforms to the journal's author guidelines.
Finally, a more philosophical but nonetheless important question: When discussing plant extraction with exotic solvents such as DES, NaDES, or ionic liquids, how do you recover the molecules of interest or the extract in dry form? While the use of NaDES can be an environmental advantage compared to using conventional solvents for this type of extraction (most often an ethanol + water mixture), the same question always arises: How do you eliminate the solvent? Or can the extract be used directly in NaDES for subsequent use in the chosen application? For the sake of objectivity, it would be beneficial to discuss this point in the Discussion section.
Once all these points have been addressed by the authors, the manuscript can, in my opinion, be definitively accepted.
Round 2
Reviewer 1 Report
Comments and Suggestions for Authors
The discussion section needs to be significantly strengthened. The results should be interpreted beyond a simple extraction efficiency and practical applicability. The investigation of the relationship between NADSE parameters and extraction performance to better understand the microenvironmental behavior of NADES.How do the current data findings advance our fundamental understanding of NADES-solute interactions?
